# The Immunomodulatory Effects of Sulforaphane in Exercise-Induced Inflammation and Oxidative Stress: A Prospective Nutraceutical

**DOI:** 10.3390/ijms25031790

**Published:** 2024-02-01

**Authors:** Ruheea Taskin Ruhee, Katsuhiko Suzuki

**Affiliations:** 1Research Fellow of Japan Society for the Promotion of Sciences, Tokyo 102-0083, Japan; 2Faculty of Sport Sciences, Waseda University, Tokorozawa 359-1192, Japan

**Keywords:** glucosinolate, sulforaphane, immunity, exercise, inflammation, oxidative stress, cytokines, reactive oxygen species, antioxidant

## Abstract

Sulforaphane (SFN) is a promising molecule for developing phytopharmaceuticals due to its potential antioxidative and anti-inflammatory effects. A plethora of research conducted in vivo and in vitro reported the beneficial effects of SFN intervention and the underlying cellular mechanisms. Since SFN is a newly identified nutraceutical in sports nutrition, only some human studies have been conducted to reflect the effects of SFN intervention in exercise-induced inflammation and oxidative stress. In this review, we briefly discussed the effects of SFN on exercise-induced inflammation and oxidative stress. We discussed human and animal studies that are related to exercise intervention and mentioned the underlying cellular signaling mechanisms. Since SFN could be used as a potential therapeutic agent, we mentioned briefly its synergistic attributes with other potential nutraceuticals that are associated with acute and chronic inflammatory conditions. Given its health-promoting effects, SFN could be a prospective nutraceutical at the forefront of sports nutrition.

## 1. Introduction

Research with bioactive compounds is getting more attention due to their unique nutritional value and numerous health benefits. Fruits, vegetables, and whole grains are considered a good source of bioactive compounds, and they have many health benefits beyond fundamental nutritional values. In addition, various epidemiological studies reported the importance of bioactive compounds in diminishing the risk of life-threatening chronic diseases, for instance, cancer, diabetes, stroke, heart disease, obesity, and so on. Some bioactive compounds are more popular with certain consumers for sustainable personalized nutrition solutions [1,2,3,4].

Our immune system consistently maintains a sustainable homogenous condition by protecting us from any harmful or foreign substances, using a generic mechanism that involves the innate or non-specific immune system [5]. This system is also known as the first-line defense system, since it reacts very quickly. The innate immune system provides an immediate response against pathogens to prevent the spread of pathogens or foreign particles throughout the body. Additionally, it involves natural killer (NK) cells, which bind to the major histocompatibility complexes (MHCs) of affected cells [6]. The specialized or adaptive immune response is also known as the second-line defense system. It works on specific types of pathogens that cause infection. The adaptive immune system is slower compared with the innate immune system, since it needs to recognize the antigen first and then function to form new antibodies to neutralize the specific antigen. This immune system has the ability to remember specific types of antigens, so that it responds quickly the next time a similar antigen is encountered. Thus, the newly formed antibodies become a permanent component of the immune system inside the body [7].

Exercise has been considered well-structured and persistent body movement to maintain physical and mental well-being properly; however, its effects on overall health and wellness depend on its intensity and duration [8]. Regular exercise can exert anti-inflammatory effects by releasing anti-inflammatory cytokines [9]. On the other hand, intense exercise may induce the augmented production of pro-inflammatory cytokines such as interleukin (IL)-6, tumor necrosis factor (TNF)-α, IL-1β, and reactive oxygen species (ROS), resulting in inflammation and oxidative stress [10,11]. Inflammation, characterized by fever, redness, pain, swelling, and dysfunction, may cause cellular proliferation and inhibition of apoptosis, consequently elevating the risk of cancer [12]. Also, oxidative stress (a disparity between the balance of oxidants (ROS) and antioxidants) occurs during the excess production of ROS that overwhelms the eliminatory effects of antioxidants, leading to a disturbance in the redox signaling and control and/or molecular damage [13]. Endogenous antioxidants work against the free radicals to neutralize the imbalanced status and maintain a sustainable physiological condition. In addition to that, food-derived natural antioxidants also promote immune functions by reducing oxidative stress [14].

GSLs (Glucosinolates) comprise a sulfur-bonded β-D-glucopyranose residue, a hydroxylamine sulfate ester, and a variable aglycon side chain that is derived from an α-amino acid (R-group). The R-group is derived from different amino acids based on which GSLs are classified as aliphatic (from alanine, leucine, isoleucine, methionine, or valine), indole (from tryptophan), and aromatic (from phenylalanine or tyrosine) [15]. GSLs were first introduced to the research community in 1831 by Robiquet and Boutrin, and around 137 GSLs have been identified up to date using modern spectroscopic methods [16,17]. GSLs are renowned secondary plant metabolites, particularly abundant in the Brassicaceae family and vegetables of the Brassica oleracea L. species, such as broccoli, cabbage, cauliflower, etc. [18]. Additionally, GSLs can be synthesized chemically, but the process is comparatively more expensive than obtaining the natural GSLs [19].

GSLs are biologically inactive and pass through enzymatic hydrolysis by a glycoprotein named myrosinase (thioglucosidase glucohydrolase) to produce a wide range of diverse biologically active substances, such as indoles, thiocyanates, isothiocyanates, and so on, with a rearrangement of their chemical structure [20]. The beneficial and adverse effects of GSLs in animal nutrition have been investigated. Nevertheless, for human nutrition, the harmful impacts of GSLs remain to be probed because of the low availability of evidence from the literature. However, the health-promoting effects of GSLs and their metabolites in humans are frequently mentioned, including immunomodulatory, cardioprotective, antibacterial, anticancer, chemopreventive, antioxidant, and anti-inflammatory functions [21]. Thus, the dietary intake of GSLs-rich foods has been identified as one of the more promising strategies to prevent or minimize inflammation and oxidative stress because of its role in activating detoxification enzymes, the scavenging of ROS, and inducing immune functions [22].

Sulforaphane (1-isothiocyanato-4-methylsulfinylbutane, SFN) is a naturally occurring isothiocyanate (ITC), which is currently the topic of active research due to its attribute as a critical regulator of cellular defenses through the activation or inactivation of vital transcription factors during any cellular stimulatory responses [23,24]. SFN could activate the nuclear factor erythroid 2-related factor 2 (Nrf2) and inhibit the nuclear factor-kappa-light-chain-enhancer of activated B cells (NF-ĸB), the major transcription factor in regulating cellular responses to inflammation and oxidative stress [20,23]. SFN, a hydrolytic product from glucoraphanin (4-methylsulfinylbutyl glucosinolate), attenuates the expression of pro-inflammatory cytokines (IL-6, TNF-α, and IL-1β) by downregulating the NF-ĸB protein expression and enhances the phase 2 enzymes such as heme oxygenase 1 (HO-1) by upregulating Nrf2, thereby reducing exercise-induced inflammation and oxidative stress [22,23]. In addition to that, SFN may mitigate exercise-induced excessive free radical production by creating an influx production of the endogenous antioxidant defense system [23]. In addition, SFN naturally induces phase 2 enzyme expression, which is important for cancer chemoprevention, and various epidemiological studies reported that sufficient absorption of SFN in the body also lowers the risk of cancer [25].

This is a brief review of the immunomodulating effects of SFN in exercise-induced inflammation and oxidative stress. Here, we summarized previous studies using SFN-rich foods or supplementation to improve immune functions after intense exercise. The potential mechanisms/mode of actions of SFN to reduce inflammation and oxidative stress due to exhaustive exercise were also illustrated. Additionally, we briefly mentioned a few of the potential therapeutic effects of SFN in various diseased conditions.

## 2. Experimental Studies with SFN Intervention

The production of free radicals and oxidants is a normal physiological process, and when we exercise, an imbalance may be created between oxidant and antioxidant production levels [26]. Most free radicals are produced in the mitochondria via the electron transport chain [27]. Endogenous antioxidants work against the free radicals to neutralize the imbalanced status and maintain a sustainable physiological condition. However, an overwhelming production of ROS may hamper normal cellular metabolic processes [28]. Any electrophilic stimulation or stress activates Nrf2; SFN could be an efficient indirect antioxidant to help an individual to recover quickly from the stressed condition [29]. Since exercise training requires lots of muscle movement, nutrient supplements before and after exercise may ease muscle fatigue or muscle pain, providing efficient exercise-induced benefits [30,31,32]. In a randomized, double-blind, placebo-controlled, cross-over designed study, it was hypothesized that SFN may be used for extended periods as a therapeutic supplement for athletes for the prevention of muscle damage, since it involves high-intensity exercise [33]. In this study, young, healthy subjects were treated with SFN (30 mg/d) for four weeks in the first trial; then, after the four-week washout period, the SFN or placebo groups were changed to the opposite treatment in the second trial. Creatine kinase (CK) and pro-inflammatory cytokine IL-6 are the two most essential markers for exercise-induced muscle damage, and long-term oral intake of SFN suppresses both markers after a single bout of vigorous resistance exercise [33]. In rodents, SFN (25 mg/kg body weight) administration for three consecutive days reduced exhaustive exercise-induced muscle damage while increasing the total antioxidant capacity and attenuating plasma lactate dehydrogenase (LDH) and CK activities [34]. Additionally, single-dose SFN (50 mg/kg body weight) administration reduced plasma LDH, glutamic oxaloacetic transaminase (GOT), and glutamic pyruvic transaminase (GPT) after a single-bout exhaustive exercise test in animals [35]. It minimized the gene expression of pro-inflammatory cytokines in the liver [35]. Pretreatment with SFN for three days (25 mg/kg body weight) also improved exercise endurance capacity [34]. Delayed-onset muscle soreness (DOMS) is a common phenomenon after eccentric exercise, while taking SFN prior to exercise may suppress DOMS after two days of eccentric exercise [36]. Exercise training positively affected endurance capacity, but SFN administration may accelerate the muscles’ antioxidant defense response, improving an individual’s running distance and duration [37]. SFN treatment ensured a safe and sound strategy to protect age-associated muscle and heart dysfunction. In an aged-mice model, it was reported that SFN-fed old mice were able to run longer than the control group [38]. In a cohort study, a 7-day intense training program supplemented with broccoli sprout juice reported that SFN mitigated several markers of oxidative stress, like the myeloperoxidase (MPO) level and lactate concentration in the blood, and improved the blood glucose profile and enhanced the physical performance and adaptation to intense exercise training [39]. Further, SFN contributed effectively in a muscular dystrophy x-linked (mdx) knock-out model; oral administration of SFN (2 mg/kg/day) for eight weeks, followed by an acute exercise protocol, protected dystrophic muscles from oxidative damage in mdx mice and improved the muscle function, ROS level, and inflammation and reduced immune cell infiltration [40]. Yang et al. conducted a study and reported that SFN protected the liver from exhaustive exercise-induced excessive ROS production [41]. We also reported that a single dose of SFN administration may protect the liver from exhaustive exercise-induced oxidative stress and inflammation [35]. So far, multiple independent studies have been conducted with SFN in both human and animal studies, and most of them reported that SFN can improve post-exercise inflammatory or oxidative stress parameters (Table 1).

## 3. How SFN Reacts within Cell Signaling Pathways

### 3.1. The Activation of Nrf2 Transcription Factor

Exercise significantly changes cellular activities in the organism while increasing oxidative stress and energetic stress. These changes must be addressed by activating or inactivating the modification of several vital transcription factors [43]. Nrf2 is an essential transcription factor that remains inactive while connected with the repressor protein Kelch-like ECH-associated protein 1 (Keap1) [44,45]. During cellular oxidative stress, Keap1 releases Nrf2 and translocates into the nucleus. It modulates gene expression by reacting with the promoter region of antioxidant-responsive elements (AREs) with the assistance of small musculoaponeurotic fibrosarcoma proteins (MAF) [45]. SFN is attributed with upregulating the expression of Nrf2-mediated phase 2 enzymes (including NADPH: quinone oxidoreductase 1 (NQO1) and heme oxygenase 1 (HO-1)) and the endogenous antioxidant enzyme gene expression (Figure 1) [35,46]. In an animal experiment using Nrf2 knock-out mice, intraperitoneal administration of SFN (25 mg/kg) reduced oxidative stress markers, i.e., TBARS and the GSSG/GSH ratio. In the Nrf2++ group, a reduced level of muscle damage markers (LDH and CK) and the downstream regulation of TBARS and the GSSG/GSH ratio lead to enhanced endurance exercise capacity from SFN-induced Nrf2 activation [42]. Changes in GSH homeostasis may be reflected in the ratio of GSSG/GSH, since the GSH concentration becomes low and the GSSG concentration is high during oxidative stress [47]. Many independent studies investigated the upsurge of the GSSG/GSH ratio due to exercise training, which may be correlated with the lactate/pyruvate ratio [48,49,50]. Monocarboxylate transporter (MCT) 1 and MCT 4 are engaged in the lactate–pyruvate interchange and metabolism. SFN pretreatment before hypoxic exercise increases the expression of the lactate transporter MCT1 and increases the running capacity, with elevated LDH activity [43]. Under stimulated conditions, Nrf2 activation may enhance the expression of MCT1 during muscle expression [51]. Bose et al. reported that an SFN diet increased the exercise capacity of old mice, which was almost similar to the young mice group fed with the non-SFN diet (regular mice diet). Additionally, SFN improved muscle strength and increased the number of stem cells with improved function in skeletal muscles. The probable mechanism was presented as the active function of Nrf2-ARE binding activity and improved skeletal muscle function in the SFN-fed old mice group. Additionally, genes involved in antioxidant, antielectrophile, and glutathione synthesis pathways play a crucial role during aging. At the same time, SFN increases the transcriptional activation of these essential genes’ expression by restoring the Keap1/Nrf2/ARE pathway [38]. Besides improving endurance capacity, SFN preintervention protects from exhaustive exercise-induced liver damage [41]. Yang et al. conducted an animal study with a mild and high dose of SFN and executed exhaustive exercise for seven consecutive days along with SFN treatment and reported that an SFN intervention improved the adequacy of antioxidative stress and reduced inflammation in the liver and therefore diminished liver damage and ameliorated exercise endurance [41].

### 3.2. The Inhibition of NF-ĸB Activity

NF-ĸB is a prime protein transcription factor that efficiently controls the expression of genes that are involved in inflammatory responses. This protein complex consists of five precursors: NF-κB1 (or p50), NF-κB2 (or p52), and RelA (or p65), RelB, and c-Rel [52]. NF-ĸB is a heterodimer that is mainly composed of either p50 or p52 and p65. The NF-ĸB subunit p50 and p52 lacks a transactivation domain. Therefore, it needs to make a complex heterodimer with a subunit of the Rel family [53]. The NF-ĸB complex is activated by two pathways: the canonical and non-canonical, or alternative, pathway. Most of the inflammatory signal is mediated through canonical pathways [54].

SFN also protects cells from inflammatory reactions by interacting with key signaling pathways and inflammatory cytokines [23]. The NF-ĸB protein is relevant to inflammatory reactions, where the inhibitor of NF-ĸB kinase (IKK) is activated in response to any stimulation. Activated IKK then phosphorylates the NF-ĸB inhibitor, IĸB, causes proteasomal degradation, leaves NF-ĸB to enter into the nucleus, and commences transcription of genes, i.e., pro-inflammatory cytokines IL-6, IL-1β, and TNF-α. These pro-inflammatory markers are also known as secondary messengers and induce the function of NF-ĸB [55]. The activation of NF-ĸB in immune cells leads to an excessive production of pro-inflammatory mediators, caused by chronic inflammatory conditions and autoimmune diseases. Both the canonical and non-canonical pathways are involved in the activation of NF-ĸB [56]. The canonical pathway is dominant at the inflammatory site and is triggered by the production of pro-inflammatory cytokines [57].

SFN is familiar in this regard due to its anti-inflammatory properties. SFN-pretreated cells interfere with NF-ĸB nuclear translocation and IĸB degradation [58]. SFN may reduce inflammation by inhibiting NF-ĸB binding to DNA [59]. Various stimuli like lipopolysaccharide, hydrogen peroxide, acrolein, and TNF-α were used against various cell lines to assess the effective dose and concentration of SFN to minimize the inflammatory responses [29,60,61,62,63,64,65]. In addition, SFN may mitigate exercise-induced endotoxin production, which triggers the production of inducible nitric oxide synthase (iNOS) and nitrate production, as well as pro-inflammatory cytokines’ gene expression [29]. Sun et al. reported that a four-week SFN treatment alleviates muscle inflammation that is attributed to Nrf2-mediated inhibition of the NF-ĸB signaling pathway [66]. Few animal studies reported that selective doses of SFN are inversely associated with inflammatory responses [67,68]. Figure 2 briefly describes the inverse association between SFN and the NF-ĸB signaling system. NF-ĸB activation also acts as a key mediator for the priming signal of NLRP3 (nucleotide-binding oligomerization domain, leucine-rich repeat, and the pyrin domain containing 3) inflammasome activation [69]. Inflammasomes are multiprotein complexes that cause inflammatory reactions [70]. Prolonged inflammation causes sepsis, since muscle tissue is more prone to damage during sepsis, which results in sepsis-related pathogenesis [71,72]. Recently, it was reported that SFN attenuated the NLRP3 protein level in muscle myoblasts and reduced the secretion of inflammatory cytokine IL-1β and toll-like receptor 4 (TLR4) [73]. Moreover, SFN treatment also restores myogenic differentiation by repressing the activation of the TLR4 pathway [73].

## 4. The Therapeutic Attributes of SFN in Combination with Other Nutrients

### 4.1. The Synergistic Effect of SFN and Other Nutraceuticals

Several clinical and preclinical studies have reported the therapeutic effects of SFN in many diseases that are related to inflammation. Subsequently, few studies have been performed on the combined effect of SFN with different nutrients like vitamin D, nobiletin (NBN), and curcumin (CUR) [74,75,76]. SFN combined with vitamin D upregulated Nrf2 expression [76]. Moreover, CUR and SFN are more effective in preventing inflammation-associated diseases. Both CUR and SFN have some efficacy to induce the Nrf2/ARE signaling pathway; however, CUR and SFN become more effective, even at a lower concentration. Cheung et al. reported that CUR and SFN synergistically induce HO-1 expression and simultaneously reduce iNOS and cyclooxygenase (COX)-2 protein expression and related inflammatory markers [74].

### 4.2. SFN as a Cancer Chemopreventive Nutraceutical

SFN is also known for significant cancer chemopreventive benefits, and a plethora of research was performed regarding the affectivity of SFN against different types of cancer like liver cancer, prostate cancer, breast cancer, ovarian cancer, pancreatic cancer, and colorectal cancer [77,78,79,80,81,82]. Cornblatt et al. performed a study to extrapolate the practical dose of SFN as a cancer chemopreventive agent and reported that SFN metabolites are readily available in the mammary tissue after receiving a single dose of SFN (200 μmole), which is prepared from myrosinase-active broccoli sprout powder [78]. This dose is equivalent to 35 mg of SFN (molecular weight of SFN: 177.29). After thirty minutes of a single dose of SFN (150 μmol), significant induction was noticed with two important cryoprotective enzymes, HO-1 and detoxification enzyme NQO1, in the mammary tissue [78]. After 12 h of SFN ingestion, a maximum of 12-fold of the induction of NQO1 was observed in the mammary tissue, while significant induction was found after 2 h. Similarly, HO-1 induction was significantly observed within one hour of ingestion [78]. Moreover, another animal study reported that SFN administration prevented tumor formation in rats who were treated with the carcinogen 9,10-dimethyl-1,2-benzanthracene [83]. SFN modulates our immune system by regulating T-cell and B-cell proliferation and phagocytic activity and by influencing the cytotoxicity in NK cells. The NK cells are critically important in controlling carcinogenesis [84]. Due to the chemopreventive effect, SFN can readily block and suppress the carcinogen [85]. Numerous mechanisms of SFN are being investigated to target multiple carcinogenetic cells. Many suggested that SFN exerted a chemopreventive function by preventing the phase 1 enzymes’ activation, along with the induction of detoxification enzymes, therefore suppressing pro-inflammatory responses within the cells [86]. The NF-ĸB signaling pathway is a critical part of the innate immune system and plays a vital role in cancer initiation and progression. The active form of NF-ĸB upregulates the anti-apoptotic gene expression, therefore, it indirectly upregulates cell proliferation [87]. In cancer patients, NF-ĸB remains active, and SFN administration may downregulate the NF-ĸB expression in prostate cancer cells [88]. Heiss et al. reported that SFN can directly inhibit the activation of the NF-ĸB subunit and reduce the DNA-binding capacity without interfering with the endotoxin-induced breakdown of the inhibitor of NF-ĸB and the nuclear translocation of NF-ĸB [59].

### 4.3. SFN and Other Chronic Diseases

Considering broccoli sprout powder (BSP) to be a rich source of SFN, a randomized, double-blind and placebo-controlled clinical trial was conducted among type 2 diabetic patients. A four-week intervention of BSP (the SFN content of BSP was determined to be ~22.5 μmol/g) resulted in the lowering of the inflammatory mediator IL-6 concentration in type 2 diabetic patients compared to the control [89]. Additionally, SFN administration (100 μmol per kg/body weight) also upregulates the insulin signaling pathway, as well as improves the glucose tolerance (Figure 3) [90]. In 2021, a clinical trial was conducted with type 2 diabetes mellitus (T2D) patients with an intervention of aerobic resistance training and broccoli supplementation (10 g/day; 22.5 mmol/g SFN) for 12 weeks [91]. They reported that broccoli supplementation with exercise training improved the lipid profile, body composition variables, and insulin level among the diabetes group compared to broccoli supplementation alone or exercise training alone [91]. Additionally, several studies reported that SFN intervention can reduce obesity through various mechanisms like the browning of fat, altering leptin resistance, and promoting lipolysis [92,93,94,95]. However, SFN showed no anti-inflammatory or antioxidative effects in patients with chronic kidney diseases (CKDs). A cross-over, randomized, double-blind study was performed with CKD patients, providing 150 μmol of SFN for two months and showing no effect in terms of Nrf2 and NF-ĸB expression or inflammatory markers [96]. An acute toxic dose of SFN (300 mg/kg body weight) causes pro-convulsion, hypothermia (150–300 mg/kg), impaired motor coordination (200–300 mg/kg), and reduced muscle strength (200–250 mg/kg) [97].

During the COVID-19 pandemic, caused by the SARS-CoV-2 (Severe Acute Respiratory Syndrome Coronavirus 2) virus, SFN was used as a drug to treat the immune cells in the lungs, which results in a reduction in T-cell activation and cytokine production [98]. Since SARS-CoV-2 directly promotes NLRP3 inflammasome activation, and SFN was reported to inhibit the inflammasome activation via Nrf2 activation, SFN may indirectly contribute to diminishing the cytokine storm in patients with COVID-19 [99,100]. However, direct studies regarding exercise and SFN intervention in COVID-19 patients are yet to be published.

## 5. Discussion and Further Perspectives

Numerous research articles have reported the interplay between SFN and its activity at the transcriptional level. In this review, we discussed mainly the impact of SFN on exercise capacity and the inherent physiological changes after an SFN intervention. SFN could be a sustainable intervention to improve exercise endurance capacity and elevate the mitochondrial function and cellular antioxidant responses. It has been reported that SFN showed its protective effect against exercise-induced ROS production with the induction of the Nrf2 pathway, which further activates several genes that are related to antioxidant and anti-inflammatory responses. SFN also protects organisms by increasing the activity of endogenous antioxidants, i.e., SOD, CAT, GPx, HO-1, and NQO1. Inflammation is a significant cause of the progression of several chronic diseases; SFN interferes with the regulation of the NF-ĸB pathway and, therefore, reduces the secretion of pro-inflammatory cytokines and other inflammatory markers’ expression. Since an excessive production of ROS impairs redox homeostasis, SFN intervention may improve the imbalanced condition by modulating several major transcription factors, like Nrf2. Moreover, SFN is thought to have an anti-inflammatory role, in conjunction with its other chemopreventive properties. This review mainly mentioned published research articles on SFN intervention and exercise outcomes. Hence, limited research articles on exercise and SFN were presented; we expanded the article with primary research on the SFN mechanisms and also briefly mentioned the therapeutic and chemopreventive role of SFN in different diseased conditions. Although various randomized control trials with different protocols were conducted and reported, a generally acceptable guideline for the intake of SFN is yet to be declared. To our knowledge, SFN is one of the most studied phytochemicals among ITCs and has shown plenty of health benefits. In order to gain a precise understanding of SFN consumption, more preclinical and clinical studies are required.

## Figures and Tables

**Figure 1 ijms-25-01790-f001:**
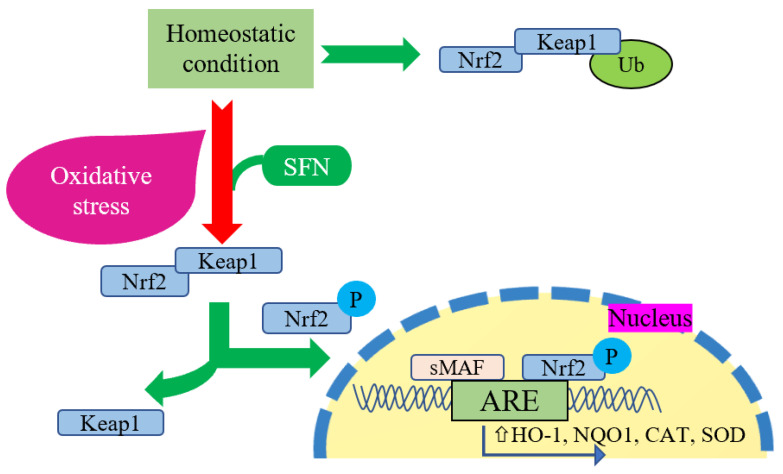
Cellular interaction between Nrf2 and SFN. Keap1 inhibits Nrf2 degradation in homeostatic conditions by promoting Nrf2 ubiquitination (Ub) in the cytoplasm. During oxidative stress, Nrf2 phosphorylates and translocates into the nucleus to promote antioxidant response element (ARE) expression, as well as induces gene expression of HO-1, NQO1, CAT, and SOD. In this regard, SFN acts as a Nrf2 activator that promotes the dissociation of Nrf2 from its negative regulator Keap1 and upregulates this reaction comprehensively.

**Figure 2 ijms-25-01790-f002:**
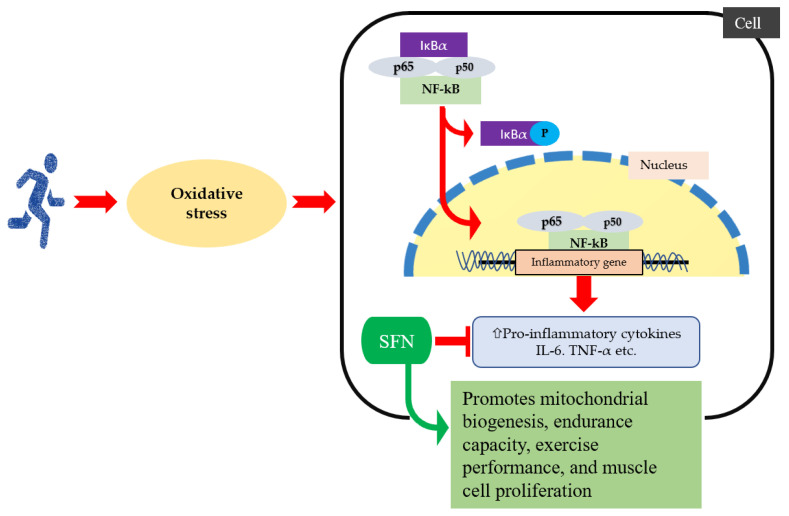
When oxidative stress increases during exercise, IκBα is phosphorylated, and an released activated NF-κB that entered into the nucleus, which increases production of pro-inflammatory cytokines and chemokine expression. SFN intervention improves exercise performance by reducing reactive oxygen species, inflammatory cytokines, and chemokines, as well as inactivated NF-ĸB signaling pathway.

**Figure 3 ijms-25-01790-f003:**
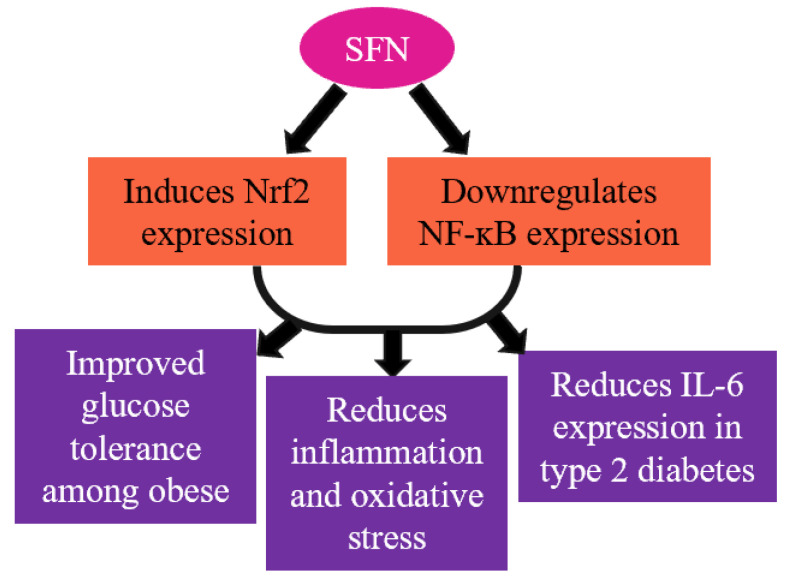
SFN mediates vital transcription factors Nrf2 and NF-ĸB and reduces inflammation and oxidative stress.

**Table 1 ijms-25-01790-t001:** Research elucidating the effect of SFN intervention on exercise-induced inflammation and oxidative stress.

Subjects	Study Design	Dose/Day, Time	Exercise Protocol	Key Findings	Reference
9 healthy adults	Randomized double-blinded cross-over	75 g broccoli sprout juice, 9 days	Intense exercise training	Improves Nrf2 expression, decreases lactate accumulation	[39]
16 healthy men	Randomized control trial	SFN tablet, 30 mg, 2 weeks	Eccentric exercise	Increases Nrf2 expression, suppresses DOMS	[36]
10 healthy men	Randomized, double-blind, placebo-controlled cross-over	SFN, 30 mg, 4 weeks	Heavy-resistance exercise	Decreases CK and IL-6	[33]
32 male Wister rats	Animal model	SFN, 25 mg/kg, 3 days	Acute exhaustive exercise (motorized treadmill)	Decreases plasma LDH and CPK, muscle MDA. Increases NQO1 and antioxidant enzyme activity.	[34]
32 male wild-type mice	Nrf2 knock-out mice model	SFN, 25 mg/kg, 3 days	Acute exhaustive exercise (motorized treadmill)	Upregulation of Nrf2 expression, reduced oxidative stress markers in skeletal muscle	[42]
12 mdx mice, 6 C57BL/10 mice	Mdx mice model	SFN, 2 mg/kg, 8 weeks	Acute exhaustive exercise (motorized treadmill)	Improved muscle function associated with Nrf2 signaling	[40]
36 male C57BL/6 mice	Animal model	SFN, 25 mg/kg, 2 h before exercise	Acute exhaustive exercise (motorized treadmill)	Reduces AST, ALT, LDH, and pro-inflammatory cytokine expression in liver through the activation of Nrf2/HO-1 signaling pathway	[35]
40 C57BL/6 mice	Cohort of old and young mice	SFN diet (442.5 mg/kg), 12 weeks	Acute exhaustive exercise (motorized treadmill)	Improved skeletal muscle function in old mice by restoring Nrf2 activity	[38]
24 C57BL/6J male wild-type mice	Animal model	SFN, 25 mg/kg, 3 days	HIIT	Improved exercise capacity by inducing Nrf2, HO-1, CAT, SOD2, and Gpx1 protein expression.	[37]
10 C57BL/6J male mice	Animal model	SFN, 10 mg/kg, 30 mg/kg, 90 mg/kg	Exercise by swimming until exhaustion	Protects liver by reducing expression of inflammatory markers and upregulating the antioxidant enzyme expression	[41]

DOMS: delayed-onset muscle soreness; CK: creatine kinase; IL: interleukin; LDH: lactate dehydrogenase; CPK: creatine phosphokinase; MDA: malondialdehyde; NQO1: NADPH quinone oxidoreductase 1; AST: aspartate aminotransferase; ALT: alanine aminotransferase; HO: heme oxygenase; HIIT: high-intensity interval training; CAT: catalase; SOD: superoxide dismutase; Gpx: glutathione peroxidase.

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
