# Peer review of "The Immunomodulatory Effects of Sulforaphane in Exercise-Induced Inflammation and Oxidative Stress: A Prospective Nutraceutical"

_ijms, 2024, doi:10.3390/ijms25031790_

Round 1

Reviewer 1 Report

Comments and Suggestions for Authors

The manuscript entitled “Immunomodulatory effects of sulforaphane in exercise- 2 induced inflammation and oxidative stress: A prospective glu- 3 cosinolates derivative” is well written and scientifically good. However there are major technical revisions should be done before acceptance.

1.     In the introduction part line number line no:69,70 broccoli (Brassica oleracea), cabbage (Brassica oleracea), cau- 69 liflower (Brassica oleracea) the scientific name of the plant should be italicized.

 2.     Author should expand iNOS and COX-2 for the first time denotion.

 3.     Authors should detailabout the biosynthesis process of SFN and also author should refer the published liteartures of doi:10.1111/bju.13361,  for SFN related biological studied and pharamalogical effects.

 4.     Authors should add a little highlights in the introduction part about use broccoli and cabbage which contains SFN as a traditional medicine for the treatment of cancer

 Reviewer recommendation: Accept after major revision.

Comments on the Quality of English Language

English revision should be done

Author Response

We are thankful to the reviewer for the valuable comments on our manuscript. We have revised the manuscript accordingly. Please see the details of the amendments which are written in italic. Thank you.

Reviewer 2 Report

Comments and Suggestions for Authors

In this manuscript, Ruhee and Suzuki studied the immunomodulatory effects of sulforaphane in exercise-induced inflammation and oxidative stress. The work is interesting, and the manuscript is well written.

Comments:

1.       In Abstract, line 19 ”we mentioned briefly its synergistic attributes…”. It is not clear if the which compound has synergistic effect with sulforaphane here. It is between to give both compounds name when you say synergistic.

2.       In the Introduction lines 30-32. “bioactive compounds in diminishing the risk of life- threatening chronic diseases, for instance, cancer, diabetes, stroke, heart disease, cataracts, Alzheimer’s, obesity, and so on”. The cataracts, Alzheimer’s, and obesity are not life- threatening chronic diseases. May be removed.

3.       Lines 304-306, Reference 89. “a clinical trial was conducted with type 2 diabetes mellitus (T2D) patients with an intervention of aerobic resistance training and broccoli supplementation (10 g/day) for 12 weeks [89]. “broccoli supplementation (10 g/day)”, it is better to give the amount of sulforaphane here.

4.       The equivalent doses of sulforaphane used in the two trials (References 87 and 88) should be given (lines 299-304) in the text.

Author Response

We are thankful to the reviewer for the valuable comments on our manuscript. We have revised the manuscript accordingly.Please see details of the amendments which are written in italic.

Reviewer 3 Report

Comments and Suggestions for Authors

This review article demonstrates the immunomodulatory effects of sulforaphane (SFN), specifically on exercise-induced inflammation and oxidative stress. SFN is one of the most exciting topics in botanical drug development. However, despite being a review article, it needs more figures or tables to summarize and present the information effectively.

How SFNs respond to cell signaling pathways, activation of Nrf2 transcription factor, inhibition of NF-ĸB activity, types of SFNs combined with other nutrients, etc. should be explained through mechanism images for each case.

In the case of Figure 2, the image needs to convey the content thoroughly. It needs to be improved.

In the case of 2 and 4, the content needs to be systematically explained by dividing the sub-content into categories. The overall description is text-oriented so that the information could be presented more effectively.

Author Response

We are thankful to the reviewers for the valuable comments on our manuscript. We have revised the manuscript accordingly.Please see details of the amendments which are written in italic.

Round 2

Reviewer 1 Report

Comments and Suggestions for Authors

The authors carried out all the necessary revisions pointed in first round of review. 

I recommend the article for publication

Author Response

Thank you for your kind assistance.

Reviewer 3 Report

Comments and Suggestions for Authors

I checked the 'author_response.pdf' and the 'revised manuscript,' the revised manuscript reflects the initial suggestions well. 

However, the first comment, "For 2 and 4, divide the sub-content into categories," does not refer to figures but to "2. Experimental studies with SFN intervention" on page 3 and "4. Therapeutic attributes of SFN in combination with other nutrients" on page 8.

Is it possible to organize the contents of '2. Experimental studies with SFN intervention' and '4. Therapeutic attributes of SFN in combination with other nutrients in the text in a more systematic way? For example, 2.1, 2.2 ...., 4.1, 4.2 ...., etc.

Author Response

We appreciate the comment and point of view here. We agree to create a subclass for both section 2 and section 4 which will make them more organize and systematic. Conceptually speaking, section 2 is exclusively focused on human and animal studies involving SFN interventions with exercise protocols. Respectfully, we think that it may not be feasible to create subclasses for this section. Indeed, we made several subclasses for section 4 and thoroughly reviewed and updated the manuscript. Please refer to the updated version of the manuscript.